# Tumour-Derived Laminin α5 (LAMA5) Promotes Colorectal Liver Metastasis Growth, Branching Angiogenesis and Notch Pathway Inhibition

**DOI:** 10.3390/cancers11050630

**Published:** 2019-05-06

**Authors:** Alex Gordon-Weeks, Su Yin Lim, Arseniy Yuzhalin, Serena Lucotti, Jenny Adriana Francisca Vermeer, Keaton Jones, Jianzhou Chen, Ruth J. Muschel

**Affiliations:** 1Nuffield Department of Surgical Sciences, University of Oxford, John Radcliffe Hospital, Oxford OX39DU, UK; 2Faculty of Medicine and Health Sciences, Department of Biomedical Sciences, Macquarie University, Sydney, NSW 2109, Australia; esther.lim@mq.edu.au; 3CRUK/MRC Oxford Institute for Radiation Oncology, University of Oxford, Oxford OX37LE, UK; yuzhalin@gmail.com (A.Y.); serena.lucotti@gmail.com (S.L.); jenny.vermeer@oncology.ox.ac.uk (J.A.F.V.); keatonjones@doctors.org.uk (K.J.); jianzhou.chen@oncology.ox.ac.uk (J.C.); ruth.muschel@oncology.ox.ac.uk (R.J.M.)

**Keywords:** liver metastasis, laminin, angiogenesis, Notch, microenvironment, matrisome

## Abstract

Hepatic metastatic growth is dependent upon stromal factors including the matrisomal proteins that make up the extracellular matrix (ECM). Laminins are ECM glycoproteins with several functions relevant to tumour progression including angiogenesis. We investigated whether metastatic colon cancer cells produce the laminins required for vascular basement membrane assembly as a mechanism for the promotion of angiogenesis and liver metastasis growth. qPCR was performed using human-specific primers to laminin chains on RNA from orthotopic human colorectal liver metastases. Laminin α5 (LAMA5) expression was inhibited in colon cancer cells using shRNA. Notch pathway gene expression was determined in endothelia from hepatic metastases. Orthotopic hepatic metastases expressed human laminin chains α5, β1 and γ1 (laminin 511), all of which are required for vascular basement membrane assembly. The expression of Laminin 511 was associated with reduced survival in several independent colorectal cancer cohorts and angiogenesis signatures or vessel density significantly correlated with LAMA5 expression. Colorectal cancer cells in culture made little LAMA5, but its levels were increased by culture in a medium conditioned by tumour-derived CD11b^+^ myeloid cells through TNFα/NFκB pathway signalling. Down-regulation of LAMA5 in cancer cells impaired liver metastatic growth and resulted in reduced intra-tumoural vessel branching and increased the expression of Notch pathway genes in metastasis-derived endothelia. This data demonstrates a mechanism whereby tumour inflammation induces LAMA5 expression in colorectal cancer cells. LAMA5 is required for the successful growth of hepatic metastases where it promotes branching angiogenesis and modulates Notch signalling.

## 1. Introduction

Although there have been significant improvements in colon cancer survivorship over the last five decades, the treatment of hepatic metastasis remains particularly challenging. Surgical resection is the only curative option. However, not all patients are suitable candidates and post-operative recurrence is commonplace.

Cells within the tumour stroma are crucial for the successful development of metastasis from primary cancers and stromal gene signatures identify colon cancers with adverse outcomes [1,2], suggesting that alterations in extracellular matrix (ECM) protein constituents affect cancer progression. In primary tumours, hypoxia leads to collagen cross-linking, resulting in the generation of a stiff, fibrotic ECM which promotes cancer cell migration and invasion [3]. Although significant differences exist in the relative abundance of different ECM proteins in primary and metastatic colon cancers as well as their normal tissue counterparts [4], the contributions of specific ECM proteins to the progression of liver metastases have not been thoroughly studied.

Laminins are glycoproteins that are abundantly expressed in the tumour extracellular matrix, first isolated from the murine Engelbreth-Holm-Swarm sarcoma [5]. They exist as heterotrimers, each composed of an α, β and γ chain, and are located primarily within the vascular and epithelial basement membranes, compartments in which their relative chain abundance differs [6]. Laminins demonstrate an organ-specific pattern of expression [7,8] and, after being deposited in the ECM, undergo enzymatic cleavage to modulate their function [9]. The globular laminin domain enables interaction with integrin receptors and other ECM proteins [10], facilitating cell adhesion and movement, the support of developing vasculature, promotion of the stem cell phenotype and collagen cross-linking. Notably, many of these functions are required for the metastatic phenotype, making laminins ideally suited to facilitate cancer progression. 

It is therefore not surprising that the over-expression of specific laminin chains is associated with cancer progression [11] and poor cancer prognosis [12,13]. Laminin γ2 expression promotes cancer cell migration and invasion and is highly expressed in budding tumour cell colonies within colorectal cancers [14] under the regulation of β-catenin [15]. Laminins 511 and 411 are key components of the vascular basement membrane [16] and are capable of self-assembly [17] with collagen IV in order to form a scaffold for endothelial cell adherence [18]. In colon cancer cells, the over-expression of laminin α1 increased angiogenesis and the growth of subcutaneous tumours [19]. In breast cancer, the expression of laminins α4 and β1 in the vascular basement membrane increased in a stepwise fashion from normal breast through primary breast cancer and distant metastasis [11]. Although tumour cells produce laminins, a mechanism whereby cancer cells contribute to the vascular basal lamina through laminin production has not been documented. Such a mechanism would enable tumour cells to produce a suitable substrate for angiogenesis and would implicate cancer-derived laminin chain expression as a novel driver of cancer angiogenesis. Nonetheless, the relevance of cancer-derived laminin and the microenvironmental mechanisms governing laminin production by tumours cells are unexplored. This is of particular relevance to the study of liver metastases which are highly angiogenic and often unresponsive to traditional antiangiogenic therapies such as those targeting the VEGF pathway [20,21,22].

Here, we demonstrate the tumour cell-specific expression of laminin chains α5, β1 and γ1 (LAMA5, LAMB1 and LAMC1 respectively) in orthotopic human liver metastases from mice and the deposition of these laminin chains within the tumour stroma. We identify LAMA5 specifically as a key promotor of liver metastasis growth and find that the inhibition of tumour-derived LAMA5 reduced branching angiogenesis in association with increased Notch signalling in tumour endothelia. This data identifies a novel mechanism through which LAMA5 expression driven by myeloid-specific inflammatory cues promotes branching angiogenesis and liver metastasis growth. Inhibiting the production of vascular basement laminins by tumour cells may serve as a useful approach to prevent the growth of hepatic metastases and the ability of tumour cells to regulate angiogenesis through the expression of vascular basement membrane laminins warrants further investigation. 

## 2. Results

### 2.1. Colon Cancer Cells in Metastatic Tumours Express Laminin 511 Which Is Associated with Adverse Colon Cancer Outcome

Immunofluorescent staining of liver metastases in mice and humans using an antibody that recognises all laminin chains of both human and murine origin demonstrated deposition in extracellular bands in both species (Figure 1A). In orthotopic liver metastases, laminin was present at high levels in the tumour ECM, where it ensheathed much of the vasculature (Figure 1B).

We next characterized the expression of all known laminin and collagen IV chains using human-specific PCR primers with RNA obtained from orthotopic liver metastases developed using three human colon cancer cell lines. Among others, human LAMA5, LAMB1 and LAMC1 transcripts were expressed at high levels, with little or no expression of human collagen IV identified (Figure 1C), indicating that the laminins required for vascular basement membrane assembly are produced by metastatic colon cancer cells. In support of the increased cancer-specific production of these laminin chains, we identified significantly greater expression of LAMA5, LAMB1 and LAMC1 transcripts in colon cancer tissues compared with normal colon in publicly available gene array data using the Oncomine^TM^ platform (Figure 1D). This was supported by the analysis of a further dataset (GSE41258) which also demonstrated an increase in laminin chain expression in hepatic metastases when compared with normal liver (Figure 1E). High levels of laminin 511 expression were also associated with reduced survival in three independent colon cancer datasets (Figure 1F), indicating the importance of these laminins for colon cancer progression.

### 2.2. Laminin 511 Expression in Primary and Metastatic Colon Cancers Is Associated with Angiogenesis

Immunohistochemical staining of orthotopic hepatic metastases and subcutaneous colon cancers in mice using an antibody specific for human LAMA5 demonstrated expression in a discontinuous pattern on the basolateral surface of cancer cells in close association with murine collagen type IV (Figure 2A). To investigate the significance of laminin 511 with respect to angiogenesis, we performed Gene Set Enrichment Analysis (GSEA) analysis using TCGA colon cancer data. We identified a striking association between the expression of a pro-angiogenic transcriptional signature [23] and the over-expression of laminin 511 (Figure 2B), indicating a correlation between high levels of laminin 511 expression and angiogenesis in colon cancer.

In primary colon cancer and liver metastasis resection specimens, LAMA5 and LAMB1 were strongly expressed within the vascular basement membrane where they encased the tumour vasculature similar to the pattern in the orthotopic tumours (Figure 2C,D). Finally, there was a significant correlation between microvessel density and LAMA5 deposition in a collection of 30 human hepatic metastases (Pearson *r* = 0.55, *p* = 0.002) (Figure 2E,F). Taken together, this data indicates that the expression of laminin 511 is associated with angiogenesis in colon cancer.

### 2.3. LAMA5 Production by Colon Cancer Cells Is Mediated by TNFα Signalling through NFκB

We next explored the potential mechanisms that regulate LAMA5 expression in colon cancer cells. We previously demonstrated that myeloid cells support liver metastasis growth through the promotion of angiogenesis [24,25,26] and so we hypothesised that inflammatory cues from these cells might promote laminin expression in colon cancer cells. In GSEA analysis, myeloid immunomes [27] were enriched in tumours over-expressing laminin 511 (Figure 3A), whereas there was no such enrichment of immunomes associated with T- or B-cell infiltration (Figure 3A), indicating that laminin 511 expression is associated with a myeloid immune infiltrate. In keeping with these findings, we identified a significant reduction in LAMA5 expression in hepatic metastases from mice depleted of neutrophils (Figure 3B,C).

To investigate these findings further, we assessed the effect of CD11b^+^ myeloid cell-conditioned media on LAMA5 expression by colon cancer cells. LAMA5 expression was increased in cells grown in media conditioned by metastasis-derived myeloid cells, while no such effect was apparent for tumour cells grown in media derived from naïve myeloid cells (Figure 3D).

Polarisation of myeloid cells within the tumour microenvironment is well recognised and a significant effect of such polarisation is the alteration of the immune cell secretome in response to microenvironmental factors. To identify potential myeloid-derived factors responsible for promoting LAMA5 expression in cancer cells, we profiled the conditioned media from naïve and liver metastasis-derived CD11b^+^ cells using protein cytokine arrays. Conditioned media from metastasis-derived CD11b^+^ cells demonstrated up-regulation of multiple cytokines including G-CSF, IL-1b, CXCL10 and TNF-α when compared with media from naïve CD11b^+^ cells (Figure 3E). The canonical NFκB pathway is downstream of TNF-α and is a major effector of TNF-α activity. G-CSF, IL-1b, and CXCL10 are released from the metastasis-derived myeloid cells and all also trigger the NFκB pathway, which plays an important role in the cancer cell response to inflammation [28]. We used TNF-α as an exemplar of NFκB-induced signalling. The treatment of colon cancer cells with TNF-α resulted in an increase in LAMA5 expression; an effect not demonstrated in endothelia (Figure 3F). 

TNF-α caused the activation of the NFκB pathway in human colon cancer cells. (Figure 3G). This was a dose-dependent increase in NFκB activation (Figure 3H) and was abrogated by addition of the NFκB inhibitor ML120B (Figure 3I). Using this system, we identified a time-dependent increase in LAMA5 expression upon TNF-α stimulation, an effect that was also abolished by co-administration of ML120B (Figure 3J). These findings link TNF-α activity and NFκB activation to the production of LAMA5 by colon cancer cells.

### 2.4. Downregulation of Tumour-Derived LAMA5 Inhibits Metastatic Colon Cancer Growth, Inhibits Branching Angiogenesis and Upregulates Notch Signalling in Endothelia

Laminins are initially secreted in their functional trimeric form [29] and inhibition of the production of a single laminin chain gene will disrupt the successful formation of all laminins containing that chain. We introduced shRNA-targeting LAMA5 (denoted LAMA5sh) to colon cancer cells which resulted in a significant reduction in LAMA5 protein expression in transfected cells (Figure 4A). The reduction in LAMA5 was associated with a significant reduction in liver metastatic burden following intrasplenic injection of colon cancer cells (Figure 4B,C). These LAMA5sh metastases contained significantly less LAMA5 expression than the controls (Figure 4D).

Given the key role that LAMA5 plays in embryonic vessel formation [30], we examined the vasculature in HT29-LAMA5sh liver metastases. LAMA5sh metastases had an equivalent CD31 staining area when compared with control metastases. However, vessels in LAMA5sh metastases were longer and had fewer branches compared to those from control metastases (Figure 4E,F). These changes are in keeping with those documented in the vascular beds of mice lacking laminin α5 [30] and suggest a role for LAMA5 in the promotion of branching angiogenesis within tumours. Interestingly, the blood vessels in LAMA5sh tumours also displayed a significant reduction in perfusion (Figure 4G,H), indicating that LAMA5 loss significantly altered functional aspects of tumour vessels as well as their morphology.

Angiogenesis is a recognised hallmark of cancer. The dysregulated angiogenesis in cancer often has excessive and irregular branching. In general, branch formation is dependent upon coordinated signalling between tip and stalk cells [31]. Notch pathway factors are crucial for determining tip or stalk cell fate in new vascular sprouts [32] and thereby regulate branching angiogenesis. To investigate the molecular mechanisms through which alterations in basement membrane LAMA5 loss might lead to alterations in vascular structure and function, we performed qPCR for a range of angiogenesis regulatory genes in endothelial cells from HT29ctrl and HT29-LAMA5sh liver metastases. Isolated endothelia expressed canonical endothelial genes CDH5, Pecam1, vWF and KDR and lacked the expression of the immune cell genes PTPRC, Itgam and Itgb2 (Figure 4i), indicating successful endothelial cell isolation. Strikingly, CD31^+^ cells from HT29-LAMA5sh tumours demonstrated the increased expression of several Notch pathway genes including *Hey2*; an essential downstream target of Notch and mediator of Notch signalling [33] (Figure 4J). CD31^+^ cells from HT29-LAMA5sh metastases also expressed higher levels of *Spry2*, a gene that inhibits branching angiogenesis through inhibition of the phospho-Erk1/2 pathway [34]. In keeping with the qPCR data in endothelia, we also identified a significant increase in *Hey2* expression in protein lysates HT29-LAMA5sh metastases (Figure 4K). 

We subsequently developed an in vitro assay to confirm the inhibition of Notch signalling in endothelia secondary TNA-α driven LAMA expression. Here, tumour cells cultured on gelatine (control HT29 or shLAMA5) are serum starved at confluence for 7 days in the presence of 100 ng/mL TNF-α or a PBS control. Following this, the plates are decellularised to leave behind an intact matrix [35]. As expected, control cells treated with TNF-α deposited LAMA5 in the matrix, whereas TNF-α did not have this effect in shLAMA5 cells (Figure 4L). The subsequent culture of 2H11 cells on HT29-derived matrices demonstrated a significant reduction in the nuclear expression of the Notch pathway proteins NICD and HEY2 (Figure 4M–O), confirming the ability of TNF-α to inhibit endothelial Notch signalling through the promotion of LAMA5 deposition. 

## 3. Discussion

Our first finding of significance is that colon cancer cells of diverse genetic backgrounds express the laminin chains found in the vascular basement membrane. Human LAMA5, LAMB1 and LAMC1 transcripts were identified in orthotopic metastases, a finding supported by immunohistochemical analysis demonstrating the expression of the human LAMA5 protein in these tumours. The expression of laminin 511 was associated with the upregulation of a set of genes regulating angiogenesis in TCGA data and, similarly, the expression of LAMA5 in hepatic metastases was associated with increased vessel density, linking the expression of these laminin chains to angiogenesis. 

Importantly, these laminin chains were located primarily within the vascular basement membrane on the basolateral surface of cancer cells, as indicated by their association with collagen IV. Colonic epithelial cells are not reported to express vascular basement membrane laminins and our data implicates the specific production of these laminin chains as a mechanism through which colonic cancer cells promote angiogenesis. Although laminin 511 expression has been studied in relation to breast cancer, the staining pattern we observe in liver metastases is different from that in breast cancers. Breast cancers demonstrated diffuse high expression, dispersed surrounding the cancer cells as well as in the vascular basement membrane. Although our findings do not exclude the possibility that additional laminin 511 is provided by endothelial, perivascular cells or fibroblasts, the profound effect on vascular morphology and function upon inhibition of LAMA5 expression specifically by colon cancer cells indicates that cancer cell laminin 511 deposition itself is indeed important for tumour angiogenesis. 

Importantly, the morphological changes we identified in vessels from LAMA5-deficient tumours are in keeping with those seen in mice lacking LAMA5, which demonstrate a deficiency in branching angiogenesis during early embryogenesis [30]. Tumour vasculature is usually highly branched, tortuous and fails to provide sufficient oxygen to the tumour microenvironment [32]; features reported to be relevant for cancer progression. Indeed, in certain settings, the hypoxia resulting from such poorly functioning vasculature promotes tumour cell invasion and cancer progression [36]. In our model, LAMA5 downregulation led to a significant reduction in vessel branching, changes that are similar to vascular normalisation [37]. However, although vascular normalisation has been reported to sensitise tumours to chemotherapy through improved tumour perfusion and thereby drug delivery [37], we found reduced perfusion upon LAMA5 loss. This may be because we analysed liver metastases at the terminal phase of their growth and that LAMA5 inhibition in fact causes vessel normalisation with a transient increase in perfusion, as is seen for the vascular normalising effects of VEGF inhibition [38]. Alternatively, vessels from LAMA5 knock-out mice demonstrate significantly reduced dilatation in response to shear stress [39] indicating the potential importance of LAMA5 in the local regulation of blood flow. Further investigation in this area is required to determine the clinical setting in which attempts to alter the expression of tumour-derived laminin may be beneficial and it will be important to ensure that the inhibition of LAMA5 does not inadvertently lead to therapeutic resistance as has been reported for VEGF inhibition in the setting of liver metastasis [40]. 

As well as identifying a significant reduction in vessel branching upon LAMA5 downregulation, we also found evidence of Notch pathway activation in endothelia from liver metastases lacking LAMA5. Branching angiogenesis is achieved through the coordinated activity of tip and stalk cells which differ in their responsiveness to VEGF-A by altering the expression of VEGFR-2 (KDR). Tip cells are found at the leading edge of branching vessels and demonstrate responsiveness to VEGF-A as well as synthesis of DLL4, a Notch ligand which activates Notch signalling in the adjacent stalk cells. The activation of Notch pathway activity in the stalk cells leads to VEGFR-2 suppression in those cells and inhibits further branching [41]. Endothelia from metastases lacking LAMA5 displayed higher expression of the key Notch target gene *Hey2* [33,42], without significant alteration in VEGFR-2 expression, suggesting that they maintain a stalk-like phenotype in keeping with reduced branching. The regulation of tip-stalk cell fate by factors in the extracellular matrix is recognised [43] but is until now unstudied in relation to LAMA5 expression. Importantly, Notch pathway activation driven by interaction between endothelial integrins and laminin has been reported in vitro [44], indicating a mechanism in which laminins regulate Notch activity through outside-in signalling. Our data firmly supports this and implicates tumour-derived LAMA5 as a regulator of endothelial Notch signalling. 

To investigate the potential stimuli that drive LAMA5 expression in colon cancer cells, we chose to analyse the role of infiltrating immune cells as we previously found them to be potent activators of angiogenesis in liver metastases [24,26] and immune cells are key regulators of tumour ECM deposition [45]. Initially, we identified the enrichment of various myeloid immunomes in tumours expressing high levels of laminin 511, whereas there was no such increase in adaptive immune cell signatures. Cancers infiltrated with myeloid cells typically display a poor prognosis [46], in keeping with our survival analysis of colon cancers expressing high levels of laminin 511 which similarly demonstrated poor outcome and enrichment of myeloid cell immunomes. Conditioned media from myeloid cells extracted from the liver metastatic microenvironment contained an array of inflammatory cytokines and promoted the expression of LAMA5 in colon cancer cells. Many of the identified cytokines are of relevance for colon cancer progression. However, TNFα was of particular interest as it can promote LAMA5 expression in human intestinal crypt cells [47]. TNFα is also relevant to liver metastasis formation as, in the setting of both colorectal and pancreatic cancer, TNFα blockade reduced metastatic outgrowth [48,49]. The metastatic outgrowth period is particularly dependent upon angiogenesis, providing further support for our findings. Whilst we could phenocopy the effect of myeloid cell-conditioned media with recombinant TNFα alone, with respect to LAMA5 in colon cancer cells, it remains to be determined whether other factors produced by tumour-derived myeloid cells can similarly drive LAMA5 expression. Nonetheless, this data demonstrates a mechanism whereby infiltrating myleloid cells drive the tumour cell expression of LAMA5 through NfKB signalling to promote angiogenesis and thereby link two hallmarks of cancer progression—inflammation and angiogenesis—to ECM protein deposition, placing the regulation of laminin expression at the centre of these processes. 

## 4. Materials and Methods

### 4.1. Cell Culture and Lentiviral Transfection

HT29, HCT-116, LoVo, MC38 and 2H-11 cells were grown in RPMI supplemented with 10% foetal calf serum under sterile, mycoplasma-free conditions. Cells were discarded after the 8th passage. Cell line authentication was performed using Short Tandem Repeat profiling (Cancer Research UK genomic facility, Leeds Institute of Molecular Medicine, March 2014). 

For inhibition of LAMA5 in colon cancer cell lines, HT29 and HCT-116 cells were transduced with lentivirus expressing GFP- and shRNA-targeting LAMA5 transcripts (Mission shRNA, Sigma Aldrich, St. Louis, MO, USA). Multiple vector constructs (5 in total) were transduced into cells and GFP^+^ cells were FACS sorted using a high-speed cell sorter (Beckman Coulter, Brea, CA, USA). LAMA5 knockdown was confirmed in transduced cell lines by Western blotting. The most successful LAMA5 knockdowns were achieved with the following sequences; CCGGACTGGATCAGGCTGACTATTTCTCGAGAAATAGTCAGCCTGATCCAGTTTTTTG and CCGGCTCGCCTCATAGGTGTCTATTCTCGAGAATAGACACCTATGAGGCGAGTTTTTG designated Sh1 and Sh2 respectively in the results section. For the generation of NFκB reporter cells, colon cancer cell lines were transduced with lentivirus expressing GFP under the control of the NFκB transcriptional response element and a puromycin resistance gene (CLS-013L, QIAGEN, Hilden, Germany). Successfully transduced cells were selected using puromycin and NFκB-GFP activity after TNFα stimulation was determined using flow cytometry. 

### 4.2. Animal Work

All studies were conducted in compliance with institutional and UK National animal welfare guidelines (UK Guidance on the operation of the Animals (Scientific Procedures) Act (2014)). For the anaesthetic, isoflurane (5% for induction, 3% maintenance) was delivered in O_2_ using a precision vaporiser. Anaesthetized, 8-week-old, female SCID mice underwent intrasplenic injection of 1 × 10^6^ colon cancer cells. Liver metastases were analysed at the 5th week following injection, as specified in our animal welfare licence. Liver metastasis volumes were calculated from ex vivo liver photos using ImageJ (https://imagej.nih.gov/ij/).

### 4.3. Immunohistochemistry

Immunohistochemistry was performed on 12-µm frozen sections from tissues embedded in Optimal Cutting Temperature compound (Thermo Scientific, Waltham, MA, USA) previously flash frozen in liquid nitrogen. Sections were fixed in ice-cold acetone before blocking in 10% serum of the secondary antibody species for 20 min. All primary antibodies were incubated for an hour at room temperature, as were fluorophore-conjugated secondary antibodies. For the primary antibodies used and their relevant concentrations, see Table 1.

Tissues were imaged using a confocal (LSM880, Zeiss) or an epifluorescent (Nikon 90i, Tokyo, Japan) microscope for four or three colours respectively. For the quantification of tumour vessel length and junction density, Angiotool software (v0.5a, Department of Health and Human Services, NIH, Bethesda, MD, USA) [50] was used. Tile scans of whole liver metastases were taken and segmented into 800 pixel^2^ images that were then loaded into the software for analysis. Using this, whole tumour cross-sections rather than representative regions were quantified. 

### 4.4. Western Blotting

Cells were lysed in RIPA buffer (ThermoFisher Scientific) on ice for 15 min before centrifuging at 15,000 rpm for 15 min at 4 °C. Extracted proteins were quantified using the Pierce BCA assay (ThermoFisher Scientific) before loading 8 ug of total protein with 10% v/v LDS sample buffer (Life Technologies, Carlsbad, CA, USA) onto a 4–12% Bis-Tris pre-cast protein gel (Sigma-Aldrich). Samples were run at 150 V on ice for approximately 1 h before transferring to nitrocellulose membranes at 40 V for 1.5 h. The membranes were incubated in 5% skim milk for 1 h before incubation overnight at 4 °C in mouse anti-human LAMA5 (CL3118, Novus Biologicals, Centennial, CO, USA), mouse anti-human *Hey2* (ab167280, ABCAM) or GAPDH (D16H11, Cell Signalling Technology, Danvers, MA, USA) diluted at 1:500 in 1% skim milk. After incubation in secondary antibody (Santa Cruz Biotechnology, Dallas, TX, USA), the membranes were incubated in ECL Prime solution (Amersham, Amersham Pl, Little Chalfont, Amersham, United Kingdon) and visualised using the Odyssey Fc imaging system (LI-COR Biosciences, Lincoln, NE, USA). Protein bands were quantified using ImageJ. 

### 4.5. Generation and Analysis of Single-Cell Suspensions from Murine Liver Metastases

Mice were injected with 199 mg/kg pentobarbitone via the intraperitoneal route. Laparotomy was performed and the inferior vena cava was accessed by way of a 29 G in-house built cannula system. The portal vein was incised and the thoracic portion of the inferior vena cava clamped with a metal clip. The liver was then perfused with PBS containing 4 mM EGTA at a rate of 5 mL/min for 1 min before perfusion with collagenase IV (Worthington Biochemical Corporation, Lakewood, NJ, USA), diluted in Hanks Balanced Salt Solution to a final concentration of 100 U/mL at 5 mL/min for 5 min. The digested liver was removed and cells were separated from the metastatic regions by dissociation between forceps. The cell suspension was then incubated in red cell lysis (BD Bioscience, San Jose, CA, USA) buffer before re-suspension in 0.5% BSA in phosphate buffered saline. 

### 4.6. Isolation of CD31^+^ Cells from Murine Hepatic Metastases

Following the generation of hepatic metastasis single-cell suspensions as described above, cells were labelled with anti-CD45 MACS antibodies and separated using magnetic columns (Miltenyi Biotech, Bergisch Gladbach, Germany) in order to remove immune cells. Following this step, the effluent was re-incubated with anti-CD31 antibodies and again separated using magnetic columns to isolate endothelial cells. 

### 4.7. CD11b^+^ Cell Isolation, Culture and Protein Expression Quantification

CD11b^+^ cells were FACS sorted from naïve murine livers, or those bearing HT29 metastases using the MoFlo XDP high-speed sorter (Beckman Coulter) and cultured for 24 h in serum-free media. The conditioned supernatant was centrifuged at 1500 rpm to remove any cellular debris. Proteome profiler membranes were incubated in the conditioned media and the assay was performed as per the manufacturer’s instructions (ARY028, R&D Systems, Minneapolis, MN, USA). Membranes were developed using the Odyssey Fc imaging system (LI-COR Biosciences, Lincoln, NE, USA) and the intensity was determined using in-house software with normalisation to positive control. 

### 4.8. qPCR

Cells were suspended in Trizol reagent (Thermo Fisher Scientific) and RNA was extracted as per the manufacturer’s instructions. RNA quality and quantity were determined using a spectrophotometer (NanoDrop 3300, Thermo Scientific). Complementary DNA was produced from 200 ng RNA using the High-capacity RNA to cDNA kit (Applied Biosystems, Foster City, CA, USA) as per the manufacturer’s instructions. cDNA was mixed with pre-designed forward and reverse KiCq Start primers (Sigma-Aldrich) in the presence of Power SYBR Green PCR Master Mix (Applied Biosystems). Samples were run on a Stratagene MX3005p PCR machine at 95 °C for 10 min followed by 40 cycles of 15 s at 95 °C and 1 min at 60 °C. Primer specificity was confirmed by performing dissociation curves for each primer pair. 

For the extraction of RNA from liver metastases, 1-mm^3^ pieces of tumour were dissociated in 500 μL Trizol reagent using a digital homogeniser (IKA). cDNA production and qPCR were carried out as described above. Laminin and collagen chains were profiled using custom gene array plates containing primers specific for human transcripts (SA Biosciences, QIAGEN). 

### 4.9. TNF and ML120B Treatment

2H-11 or human colon cancer cell lines were grown to 70% confluence before incubation in media supplemented with 1% FCS for 24 h. Cells were treated with the NFκB inhibitor ML120B (R&D systems) at a final concentration of 20 nM, incubated for 90 min, then recombinant murine or human TNFα (both R&D systems) was added at the indicated concentrations. The specificity of ML120B as an inhibitor of IkappaB Kinase beta (Iκκβ) inhibitor and therefore TNFα-driven NFκB activation has previously been demonstrated [51]. Cells were grown in the presence of TNFα for variable time periods prior to RNA extraction. 

### 4.10. Tumour-Derived Matrix Generation

HT29 cells were cultured until confluence on angiogenesis slide chambers (Ibidi) coated with 0.2% cross-linked gelatine. At this point, the serum-containing medium was changed for serum-free RPMI. After 24 h, the medium was supplemented with a new serum-free medium containing 100 ng/mL TNF-α or a PBS control. Supplemented media were changed every 24 h. After 7 days, all media were removed, plates were washed in cold PBS and cells incubated in a decellularisation solution containing 0.5% (v/v) Triton X-100 and 20 mM NH4OH in PBS. After 5 min, the chambers were gently washed with PBS three times and then incubated with PBS supplemented with 10 U/mL DNase overnight. At this point, 2H11 endothelial cells were plated and cultured for 48 h prior to immunocytochemical analysis. 

### 4.11. Immunocytochemistry

Endothelial cells were fixed/permeabilised in 2% paraformaldehyde with 0.1% Triton X100 for 10 min prior to blocking with 10% serum of the secondary antibody species for 20 min. All primary antibodies were incubated overnight at 4 °C prior to incubation with fluorochrome conjugated secondary antibodies for 1 h at room temperature. Chambers were visualized using a laser scanning confocal microscope (LSM880, Zeiss, Carl-Zeiss-Straße, Oberkochen, Germany). Median fluorescence Intensity (MFI) of nuclear staining in cultured endothelial cells was determined using image J. Nuclei were identified as DAPI-positive regions and fluorophore staining in these areas determined using inbuilt algorithms within Image J. 4.11. Bioinformatics (Broad Institute of MIT, Cambridge, MA, USA).

The meta-analysis of multiple colon cancer datasets comparing laminin mRNA chain expression in colon cancer and normal colon was performed using Oncomine^TM^ (Waltham, MA, USA). Gene Expression Omnibus (GEO) datasets (GSE41258 and GSE39582) and the TCGA dataset used to generate survival curves in colon cancer patients were accessed using GEO2R (National Center for Biotechnology Information) and cBioportal [52,53] respectively. Z-scores were generated from raw gene expression data and the over-expression of LAMA5, LAMB1 or LAMC1. For each gene, the Z-score identifying the upper quartile of patients was used as a cut off. Patients were considered to have a positive gene signature if their tumours expressed either LAMA5, LAMB1 or LAMC1 above the Z-score cut off. These patients were then compared with the remaining patients in the dataset. 

Gene Set Enrichment Analysis (GSEA) was performed using the Broad Institute desktop application (http://software.broadinstitute.org/gsea/index.jsp) [54]. Hand-curated .gct, .cls and .gmx files were generated using relevant GSE colon cancer datasets. Genesets for angiogenesis and cancer immunomes were taken from publications by Masiero et al. [23] and Bindea et al. [27] respectively. 

### 4.12. Statistics

Statistical analysis was performed using GraphPad Prism V7 (San Diego, CA, USA). Data is presented as the mean and standard error of the mean unless otherwise stated. Students *t*-test was used to determine differences between groups of normally distributed data with *p* ≤ 0.05 considered significant. Differences in mRNA gene expression between experimental and control groups were determined using the ΔΔct method and β-actin was used as the housekeeping gene throughout. Spearman’s correlation coefficient was used to calculate the relationship between laminin deposition and vascularity in human liver metastasis specimens. Kaplan–Meier curves were used to generate survival estimates which were then compared using the Log-rank test. For the analysis of laminin gene expression in colon cancers defined by angiogenic gene signature or the cancer immunome, the false discovery rate was reduced by applying the Benjamini–Hochberg procedure and these calculations were performed in cBioportal. Throughout the figures * *p* = 0.01 to ≤ 0.05, ** *p* = 0.001 to ≤ 0.01 and *** *p* ≤ 0.001. Animal experiments were repeated in triplicate. 

## 5. Conclusions

This work demonstrates a novel mechanism whereby colon cancer cells stimulated by inflammatory mediators produce vascular basement membrane laminins. These laminins are in turn required for successful metastatic growth and promote the development of a highly branched vascular network. Targeting laminin expression in the tumour microenvironment may serve as a novel way of reducing branching angiogenesis and preventing cancer progression. 

## Figures and Tables

**Figure 1 cancers-11-00630-f001:**
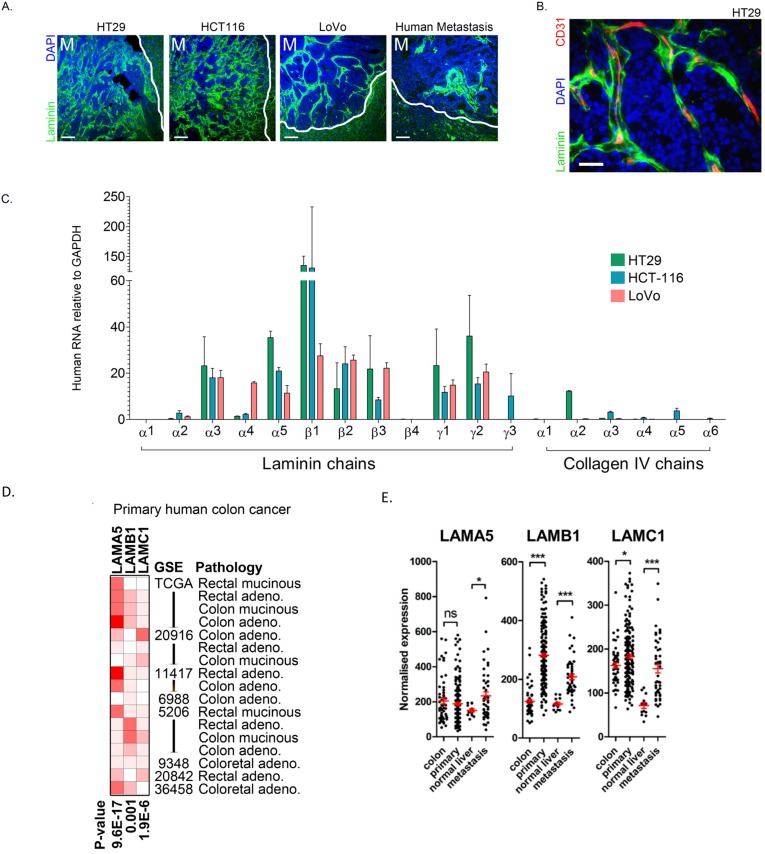
Laminin 511 is expressed by colon cancer cells in metastatic tumours and is associated with adverse outcomes in colorectal cancer. (**A**) Immunofluorescence staining for laminin (green) in orthotopic liver metastases developed using HT29, HCT116 or LoVo human colon cancer cell lines or in a human liver metastasis resection specimen. Counterstained with DAPI. M indicates metastasis and the white border demarcates the metastasis–liver interface. Scale bar represents 100 μm. (**B**) Immunofluorescence staining of HT29 orthotopic liver metastasis stained for the indicated markers using non species-specific antibodies. Scale bar represents 30 μm. (**C**) mRNA profile of all known laminin and collagen type IV chain genes in RNA extracted from orthotopic HT29 (green), HCT116 (blue) and LoVo (pink) liver metastases using human-specific primers. (**D**) Oncomine^TM^ data demonstrating mRNA expression of LAMA5, LAMB1 and LAMC1 in colon cancer relative to normal colon from multiple independent GSE datasets. (**E**) Expression of the indicated laminin chains in normal colon, primary colon cancer, normal liver and liver metastases through analysis of data from GSE41258. Error bars are the SEMs compared with Students *t*-test. (**F**) Kaplan–Meier curves (Log-rank comparison) showing cancer-specific or absolute survival in data generated from publicly available gene array datasets. Patient groups are defined by over-expression of LAMA5, LAMB1 and LAMC1 (red) or normal expression of these genes (blue). * 0.01 < *p* ≤ 0.05, *** *p* ≤ 0.001.

**Figure 2 cancers-11-00630-f002:**
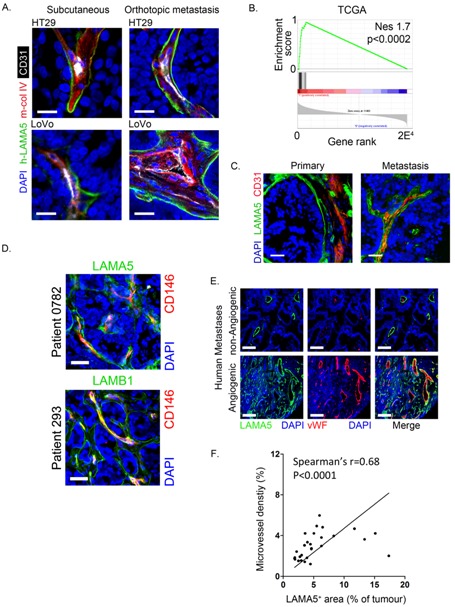
Laminin 511 expression in primary and metastatic colon cancers is associated with angiogenesis. (**A**) Immunofluorescence staining of HT29 (top) and LoVo (bottom) orthotopic liver metastases and subcutaneous tumours stained for LAMA5 (human-specific antibody, h-LAMA5, green), collagen type IV (red) and CD31 (white). Scale bars represent 20 μm. (**B**) Gene Set Enrichment Analysis (GSEA) for enrichment of a 20-gene signature regulating angiogenesis in TCGA colon cancers over-expressing laminin LAMA5, LAMB1 and LAMC. (**C**) Immunofluorescence staining of archival human primary colon cancer and liver metastasis resection specimens stained for LAMA5 (green) and CD31 (red). Scale bars represent 40 μm. (**D**) Immunofluorescence staining of archival liver metastasis resection specimens stained for LAMA5 and LAMB1 (both green) and CD31 (red). Scale bars represent 40 μm. (**E**) Immunofluorescence staining of human liver metastases from a patient with low microvessel density (top row) and one with high microvessel density (lower row) stained for LAMA5 (green) and von Willebrand factor (red). Scale bars represent 100 μm. (**F**) Scatter plot of the percentage of liver metastasis area staining positive for LAMA5 (x) vs microvessel density (y) determined by measuring the percentage area of the tumour staining positive for von Willebrand factor. Data shown for liver metastases from 30 patients with at least five tumour areas quantified per patient (Spearman’s rank correlation).

**Figure 3 cancers-11-00630-f003:**
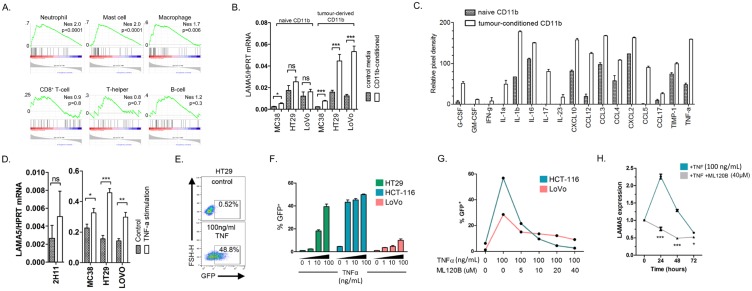
Laminin 511 expression is promoted by myeloid cell activity. (**A**) GSEA for enrichment of gene signatures reported to define specific immune cell infiltrates (immunomes) in TCGA colon cancers over-expressing LAMA5, LAMB1 and LAMC1. (**B**) Immunofluorescence staining for LAMA5 in HT29 liver metastases from control mice (top) and those treated with anti-Ly6G (1A8), an antibody that depletes neutrophils (lower). Scale bar represents 20 μm. (**C**) Quantification of the percentage of the liver metastatic area staining positive for LAMA5 in HT29 or HCT116 tumour-bearing control mice (green), or those depleted of neutrophils (pink). A minimum of five mice per group were included with at least three tumour areas quantified per mouse. (**D**) LAMA5 expression in cultured murine and human colon cancer cells after 24 h of culture in naïve or liver metastasis-derived CD11b^+^ myeloid cell-conditioned media. Error bars generated from biological triplicates. (**E**) Semi-quantitative analysis of the concentration of various cytokine proteins in the conditioned media from naïve (grey) or liver metastasis-derived (clear) CD11b^+^ myeloid cells determined using a commercially available antibody-based protein array. Error bars generated from biological triplicates. (**F**) LAMA5 expression in cultured murine and human colon cancer cells or the murine endothelial cell line 2H11, 24 h after stimulation with PBS (grey) or 100 ng/mL TNFα (clear). Error bars generated from biological triplicates. (**G**) Flow cytometry of NFκB-GFP reporter HT29 cells 24 h following treatment with PBS (top) or 100 ng/mL TNFα (below) demonstrating an increase in GFP expression in the later. (**H**) Percentage GFP^+^ HT29, HCT116 or LoVo cells 24 h following treatment with increasing concentrations of TNFα to a maximum concentration of 100 ng/mL. Error bars generated from biological triplicates. (**I**) Percentage GFP^+^ HCT116 or LoVo cells 24 h following treatment with TNFα and/or the NFκB inhibitor ML120B as indicated. (**J**) LAMA5 expression in HCT116 cells treated with 100 ng/mL TNFα with (grey) or without (green) 40 μm ML120B. Error bars generated from biological triplicates. Student’s *t*-test throughout. *p* < 0.05 considered significant. * 0.01 < *p* ≤ 0.05, ** 0.001 < *p* ≤ 0.01, *** *p* ≤ 0.001, ns: not significant.

**Figure 4 cancers-11-00630-f004:**
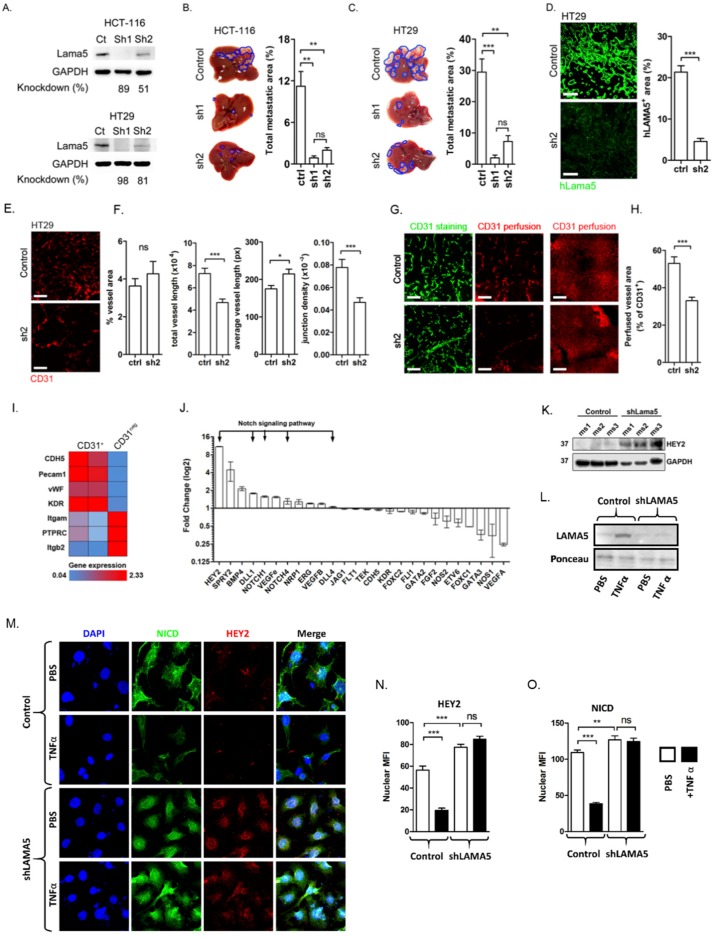
Inhibition of tumour-derived LAMA5 inhibits the growth of colorectal liver metastases and promotes endothelial Notch signalling. (**A**) Western blotting for LAMA5 in Hct-116 and HT29 colon cancer cell lines transfected with non-targeting control (ctrl) or one of two LAMA5 shRNA constructs (sh1 or sh2). (**B**) Liver metastasis growth in Hct-116 ctrl and Hct-116-sh1 or Hct-116-sh2 cell lines. (**C**) Liver metastasis growth in HT29ctrl and HT29-sh1 or HT29-sh2 cell lines. (**D**) Immunofluorescence staining for the human LAMA5 protein in liver metastases developed using HT29ctrl or HT29-sh2 cell lines. Scale bars represent 40 µm. (**E**) Immunofluorescence staining for CD31 in liver metastases developed using HT29ctrl or HT29-sh2 cell lines. Scale bar represents 40 µm. (**F**) Quantification of total CD31^+^ area, total and average vessel length and vessel junction density in liver metastases developed using HT29ctrl or HT29-sh2 cell lines. (**G**) Immunofluorescence staining for vessel perfusion (tail vein injection of anti-CD31) in red and endothelial cells (immunohistochemistry for CD31) in green in HT29ctrl and HT29-sh2 liver metastases. Scale bars represent 40 µm. (**H**) Quantification of the percentage of perfused vessels in liver metastases developed using HT29ctrl and HT29-sh2 cell lines. (**I**) RNA expression of endothelial and immune cell genes in endothelial (CD31^+^) and non-endothelial (CD31^neg^) cells MACS-separated from liver metastases generated in mice using HT29ctrl or HT29-sh2 cell lines. (**J**) Endothelial cell expression of genes involved in the regulation of angiogenesis presented as fold-change in expression in CD31^+^ cells MACS-separated from HT29-sh2 metastases relative to those from HT29ctrl metastases. (**K**) Western blotting for Hey2 protein in lysates derived from hepatic metastases developed using HT29ctrl or HT29-sh2 cell lines. (**L**) Blotting for LAMA5 in matrices derived from HT29 control and shLAMA5 cells following 7-day serum-starved culture with and without TNFα supplementation at 100 ng/mL. (**M**) Representative immunocytochemistry images of 2H11 endothelia following culture on matrices from (**L**) stained for the indicated Notch pathway proteins. (**N**,**O**) Quantification of HEY2 and NCID nuclear staining intensity in endothelia from (**M**). Analysis performed on at least 40 cells across experiments in triplicate. Student’s *t*-test throughout. *p* < 0.05 considered significant. * 0.01 < *p* ≤ 0.05, ** 0.001 < *p* ≤ 0.01, *** *p* ≤ 0.001, ns: not significant.

**Table 1 cancers-11-00630-t001:** Immunohistochemistry/Immunocytochemistry antibodies.

Antigen	Company	Species Specificity	Clone	Dilution
Laminin	ABCAM	Human, mouse	AB11575	1:200
Laminin α5	Millipore	Human	4C7	1:200
Laminin β1	Millipore	Human	4E10	1:200
Laminin γ1	Santa Cruz Biotechnologies	Human, mouse	D-3	1:200
Collagen IV	ABCAM	Mouse	AB19808	1:200
CD31	ABCAM	Mouse, human	MEC 7.46	1:100
Von Willebrand factor	ABCAM	Human	AB6994	1:100
NCID	ABCAM	Human/mouse	n/a	1:500
HEY2	ABCAM	Human/mouse	n/a	1:500

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
