# Peer review of "Tumour-Derived Laminin α5 (LAMA5) Promotes Colorectal Liver Metastasis Growth, Branching Angiogenesis and Notch Pathway Inhibition"

_cancers, 2019, doi:10.3390/cancers11050630_

Round 1

Reviewer 1 Report

The authors have adequately addressed all my comments, thus I endorse publication of the manuscript.

Reviewer 2 Report

The authors adequately addressed the raised comments, added experimental evidence supporting the hypothesis and thoroughly edited the manuscript.

This manuscript is a resubmission of an earlier submission. The following is a list of the peer review reports and author responses from that submission.

Round 1

Reviewer 1 Report

In this manuscript Gordon-Weeks et al. aim to investigate the role of tumor-derived lamin α5 in colorectal cancer liver metastasis, angiogenesis and regulation of signaling pathways involved in these processes. The authors report that laminin chains α5, β1 and γ1 are expressed in the stroma of liver metastases derived from orthotopic implantation of colorectal cancer cells in mouse models and high laminin expression is associated with worse survival of patients. They also demonstrate that laminin 511 expression is induced by myeloid cells and that laminin α5 plays key role in promoting growth of liver metastases since shRNA-mediated depletion of LAMA5 suppresses angiogenesis and induces Notch signaling in tumor endothelial cells. Overall, these findings are potentially interesting, however, some important issues remain to be addressed prior to considering publication of this manuscript.

Major comments:

1.       In Figure 1, the authors show immunofluorescene analysis of laminin expression in samples from human liver metastases. It would be interesting to perform similar analysis in primary human colorectal tumors to investigate whether laminin expression is higher in metastatic compared to primary tumors. To this end, quantification of these image would also be important

2.       The authors conclude that depletion of LAMA5 suppresses branching of angiogenesis and also vessel perfusion (Figure 4G). They also correctly discuss that this appears to be contradictory with existing literature regarding vascular normalization and vessel perfusion. However, their methodology used to inject anti-CD31 via the tail-vein may not be the most appropriate to measure perfusion and therefore results are most likely misleading. Rather, intracardiac injection of biotinylated lectin followed by IHC analysis of lectin and CD31 is one of the most established methods for measuring perfusion ex vivo. Experiments should be repeated using this methodology (i.e. see Papageorgis et al., Sci Rep. 2017).

3.       The authors demonstrate that TNFα mediates, at least in part, LAMA5 upregulation. The clinical significance of these findings would greatly benefit from the use of a TNFα inhibitor to assess its effect on LAMA5 in vitro and most importantly on the formation of liver metastases in vivo.

Minor comments:

1.       In Fig. 1E, the authors should more clearly explain how patients were stratified in terms of their expression for LAMA5, LAMB1 and LAMC1. Was the mean expression of all 3 genes used? Also, the authors mention that high expression group included z score > 1.2.  Why was this cut-off used and how did this separate the 2 patient groups (i.e. upper/lower tertile or quartile etc).

Author Response

In this manuscript Gordon-Weeks et al. aim to investigate the role of tumor-derived lamin α5 in colorectal cancer liver metastasis, angiogenesis and regulation of signaling pathways involved in these processes. The authors report that laminin chains α5, β1 and γ1 are expressed in the stroma of liver metastases derived from orthotopic implantation of colorectal cancer cells in mouse models and high laminin expression is associated with worse survival of patients. They also demonstrate that laminin 511 expression is induced by myeloid cells and that laminin α5 plays key role in promoting growth of liver metastases since shRNA-mediated depletion of LAMA5 suppresses angiogenesis and induces Notch signaling in tumor endothelial cells. Overall, these findings are potentially interesting, however, some important issues remain to be addressed prior to considering publication of this manuscript.

Major comments:

1.      In Figure 1, the authors show immunofluorescene analysis of laminin expression in samples from human liver metastases. It would be interesting to perform similar analysis in primary human colorectal tumors to investigate whether laminin expression is higher in metastatic compared to primary tumors. To this end, quantification of these image would also be important

Response: We agree that the comparison of primary and metastatic colorectal cancers would be of interest.  The best way to make biologically relevant inferences from this approach would be to compare matched primary and metastatic tumours from the same patients.  Currently we do not have access to such samples for histological analysis or a large enough bank of un-matched primary and metastatic tumours to make statistically rigorous comparisons. 

However, to address this question we have analysed the GSE41258 RNA sequencing dataset, in which expression profiling of normal colon and matched primary colon cancer, and normal liver and matched liver metastasis were performed (See Figure 1E). Here, whilst there is no increase in expression of the laminin chains when comparing primary and metastatic tumours, there is a significant increase in expression when comparing normal colon with colon cancer or normal liver with liver metastasis. This suggests that a step-wise increase in laminin expression is not required for the progression from primary cancer to liver metastasis and that the acquisition of laminin expression occurs during the development of primary colorectal cancer. This has been documented in the text (lines 116-118).

Note that in our original manuscript (Figure 2c), we stained primary human colorectal cancer for LAMA5 and identified it in the same location within the vascular basement membrane as for hepatic metastases. 

2.       The authors conclude that depletion of LAMA5 suppresses branching of angiogenesis and also vessel perfusion (Figure 4G). They also correctly discuss that this appears to be contradictory with existing literature regarding vascular normalization and vessel perfusion. However, their methodology used to inject anti-CD31 via the tail-vein may not be the most appropriate to measure perfusion and therefore results are most likely misleading. Rather, intracardiac injection of biotinylated lectin followed by IHC analysis of lectin and CD31 is one of the most established methods for measuring perfusion ex vivo. Experiments should be repeated using this methodology (i.e. see Papageorgis et al., Sci Rep. 2017).

Response: There are various methods to determine vessel perfusion of which tail-vein injection of fluorescently labelled CD31 antibody is one that has been used in various peer reviewed papers (see PMID 28289267 for example).  The liver is a highly vascular organ receiving 20-25% of the cardiac output as well as significant portal blood.  As a result, the rate limiting factor in analysis of tumour perfusion within the liver is unlikely to be technical problems caused by poor delivery of labelled antibody.  This is exemplified by the fact that in Figure 4G we see equal and adequate perfusion in the un-involved livers of mice bearing control or LAMA5sh (shRNA targeting LAMA5) metastases.  This indicates that the technique successfully detects perfused tissues as the antibody was able to reach vascular beds adequately.  

We have given several plausible reasons why we saw morphological evidence of vascular normalisation (reduced branching and vessel elongation) but reduced perfusion in response to LAMA5 loss.  The most likely reason is that vascular normalisation occurs for a short time ‘window’ which ends at the point at which vascular loss starts to occur.  Clearly it would be interesting to examine these findings in further detail but we do not feel that repeating the murine studies with a different method of vessel assessment would logically lead to a different result and further investigation of the links between LAMA5 inhibition and vascular normalisation are outside of the scope of the current study. 

3.       The authors demonstrate that TNFα mediates, at least in part, LAMA5 upregulation. The clinical significance of these findings would greatly benefit from the use of a TNFα inhibitor to assess its effect on LAMA5 in vitro and most importantly on the formation of liver metastases in vivo. 

Response: The effect of TNFα in promoting liver metastasis outgrowth through promotion of an inflammatory tumour microenvironment has previously been documented using murine models of liver metastasis where TNFα inhibition using clinically relevant therapeutics such as Infliximab inhibited metastatic growth (see PMIDs 24397824 and 18316608).  Interestingly, in these studies anti-TNF therapy inhibited the outgrowth phase of colorectal hepatic metastases during which angiogenesis is an important determinant – further supporting the link between TNF-driven laminin production. These findings have been discussed and the references added to the revised manuscript.

In the revised manuscript we have strengthened the link between TNFα activity on cancer cells, LAMA5 expression and endothelial Notch pathway regulation using an in-vitro assay (Figure 4L-M revised manuscript).  Here, tumour-cell conditioned gelatin was developed over long-term culture, allowing laminin-rich matrix deposition followed by removal of tumour cells through decellularisation (see PMID 27245425 for method).  In keeping with a mechanism whereby TNF-α-driven LAMA5 expression by tumour cells de-regulates Notch signalling in endothelia, we demonstrate a reduction in Notch activation (nuclear NICD and SPRY2 expression) following treatment of cancer cells with TNFα; an effect that was not seen in endothelia grown on matrices derived from LAMA5-deficient cancer cells. This result thereby provides confirmation of the role that tumour-derived LAMA5 expression through TNFα stimulation inhibits Notch activity in endothelial cells.

Minor comments:

1.       In Fig. 1E, the authors should more clearly explain how patients were stratified in terms of their expression for LAMA5, LAMB1 and LAMC1. Was the mean expression of all 3 genes used? Also, the authors mention that high expression group included z score > 1.2.  Why was this cut-off used and how did this separate the 2 patient groups (i.e. upper/lower tertile or quartile etc).

Response: We agree that our methodology was unclear and that the use of a z-score cut off was not properly justified.  The survival curve data has now been re-analysed with this in mind and new curves presented. This does not significantly alter the conclusions however it should be noted that there is no longer a disease-specific survival difference in the TCGA cohort (See Figure 1E).

For the current analysis, z-scores were derived for each gene (LAMA5, LAMB1, LAMC1). The z-score identifying the upper quartile of patients for each gene (analysed independently) was then determined and patients with tumours expressing one or more of the three genes at a level in the upper quartile were defined as having high signature expression. All other patients were defined as the control cohort. Comparison of the resultant survival curves was then performed.

This information has now been added to the methods section for clarity.

Reviewer 2 Report

The references need fixing.

The correlation between laminin-α5 deposition and micro-vessel density should be analyzed by Spearman's correlation.

The immunofluorescence images in Figure 3B are not convincing. It is not clear how the scoring of laminins staining was performed.

In Figure 3H, statistics and the concentrations of TNFα should be in the figure.

The tests used in statistical analysis should be included in the figure legends.

The specificity of ML120B as an NFkB inhibitor was not shown.

Indicate the sequences of the shRNA targeting laminin-α5.

The correlation with Notch pathway is weak and not much evidence of the involvement is provided. The authors should provide more mechanistic proof of the involvement of notch pathway in laminin signaling in vitro and in vivo.

Author Response

The references need fixing.

Response: We have removed any doi numbers that were included in references and have changed the reference style to the journal’s (MDPI) format.

The correlation between laminin-α5 deposition and micro-vessel density should be analyzed by Spearman's correlation.

Response: This has now been done. The findings have remained essentially the same and the new analysis is now presented in the revised manuscript (Figure 2F).

The immunofluorescence images in Figure 3B are not convincing. It is not clear how the scoring of laminins staining was performed.

Response: We agree with the reviewer and have now removed panels B and C from Figure 3 (previously containing the immunofluorescence images).  The exclusion of these panels does not alter the over-arching argument of Figure 3, that inflammatory cues drive laminin expression in colon cancer cells.

In Figure 3H, statistics and the concentrations of TNFα should be in the figure.

Response: These have now been added.

The tests used in statistical analysis should be included in the figure legends.

Response: The figure legends have been altered throughout to take account of this comment. 

The specificity of ML120B as an NFkB inhibitor was not shown.

Response: The specificity of ML120B for the IKKβ protein and therefore NFkβ pathway inhibition over other kinases has previously been demonstrated (see PMID 16525037).  We have now cited this reference in our manuscript as evidence for the specificity of ML120B. 

Indicate the sequences of the shRNA targeting laminin-α5.

Response: The sequences have been added to the methods section.

The correlation with Notch pathway is weak and not much evidence of the involvement is provided. The authors should provide more mechanistic proof of the involvement of notch pathway in laminin signaling in vitro and in vivo.

Response: To further support and strengthen the in-vivo work demonstrating up-regulation of Notch pathway transcripts in the endothelia from tumours lacking LAMA5, we have utilised an in-vitro culture system to enable comparison of the effects of tumour-derived LAMA5 on endothelial Notch signalling (Figure 4L-O revised manuscript).

Here, HT29 control or LAMA5sh cells were cultured in control, serum depleted media or media supplemented with TNFα on gelatin to promote matrix deposition (see PMID 27245425 for method on which this assay is devised).  After a week, cultures were decellularised leaving the deposited matrix and gelatin. Endothelial cells can then be grown on the decellularised matrix and analysed for Notch pathway expression.

As expected, we found higher deposition of LAMA5 when endothelial cells were grown on TNFα-treated HT29 matrices compared to those from serum starved control or shLAMA5 cells (Figure 4L).

Immunocytochemical analysis of endothelial cells demonstrated inhibition of Notch pathway proteins NCID and HEY2 when grown on LAMA5-rich matrices from TNFα treated control cells when compared with culture on matrices from the shLAMA5 cells or untreated HT29 cells (Figure 4M-O).

This provides confirmation that tumour-derived LAMA5 expression through TNFα stimulation inhibits Notch activity in endothelial cells, supporting our findings in hepatic metastases.

This experiment has been detailed in the methods section  and discussed in the result